# Predictive Factors and Clinical Impacts of Delayed Isolation of Tuberculosis during Hospital Admission

**DOI:** 10.3390/jcm12041361

**Published:** 2023-02-08

**Authors:** Inhan Lee, Soyoung Kang, Bumsik Chin, Joon-Sung Joh, Ina Jeong, Junghyun Kim, Joohae Kim, Ji Yeon Lee

**Affiliations:** 1Division of Pulmonary and Critical Care Medicine, Department of Internal Medicine, National Medical Center, Seoul 04564, Republic of Korea; 2Department of Infection Control and Prevention, National Medical Center, Seoul 04564, Republic of Korea; 3Division of Infectious Diseases, Department of Internal Medicine, National Medical Center, Seoul 04564, Republic of Korea; 4Division of Pulmonary, Allergy and Critical Care Medicine, Department of Internal Medicine, Hallym University Dongtan Sacred Heart Hospital, Hallym University College of Medicine, Hwaseong 18450, Republic of Korea

**Keywords:** tuberculosis, isolation, contact investigation, infection control, health personnel, occupational exposure

## Abstract

Delayed isolation of tuberculosis (TB) can cause unexpected exposure of healthcare workers (HCWs). This study identified the predictive factors and clinical impact of delayed isolation. We retrospectively reviewed the electronic medical records of index patients and HCWs who underwent contact investigation after TB exposure during hospitalization at the National Medical Center, between January 2018 and July 2021. Among the 25 index patients, 23 (92.0%) were diagnosed with TB based on the molecular assay, and 18 (72.0%) had a negative acid-fast bacilli smear. Sixteen (64.0%) patients were hospitalized via the emergency room, and 18 (72.0%) were admitted to a non-pulmonology/infectious disease department. According to the patterns of delayed isolation, patients were classified into five categories. Among 157 close-contact events in 125 HCWs, 75 (47.8%) occurred in Category A. Twenty-five (20%) HCWs had multiple TB exposures (*n* = 57 events), of whom 37 (64.9%) belonged to Category A (missed during emergency situations). After contact tracing, latent TB infection was diagnosed in one (1.2%) HCW in Category A, who was exposed during intubation. Delayed isolation and TB exposure mostly occurred during pre-admission in emergency situations. Effective TB screening and infection control are necessary to protect HCWs, especially those who routinely contact new patients in high-risk departments.

## 1. Introduction

Tuberculosis (TB) remains a major public health problem worldwide and was the primary cause of death from a single infectious agent before coronavirus disease (COVID-19). According to the World Health Organization (WHO), 7.1 million patients were newly diagnosed with TB in 2019, and 1.2 million human immunodeficiency virus (HIV)-negative patients died from TB [1]. After the outbreak of COVID-19, the TB mortality rates increased further, resulting in 1.3 million deaths of HIV-negative patients with TB due to reduced access to TB diagnosis and treatment [1].

In 2021, the incidence rate of TB in South Korea was 44.6 cases per 100,000 population, with 22,904 cases in total [2]. Although the number declined with government effort, the incidence rate was the highest and mortality ranked the third highest among the 38 Organization for Economic Cooperation and Development member countries [1].

Delayed diagnosis of TB is an important issue in healthcare settings, because it increases the morbidity and mortality rates of patients and in-hospital transmission of TB [3,4,5]. The WHO reported that the risk of TB infection in healthcare workers (HCWs) was 2–10 times higher than that in the general population [6]. In-hospital transmission and predisposing factors have been studied to improve HCW protection. De Vrie et al. found that delayed diagnosis of index cases, especially in older patients with comorbidity, was the main cause of nosocomial TB transmission among HCWs [7]. Yutaka et al. suggested that a longer duration of contact, higher mycobacterial load in sputum, and diagnosis in the ward are significant risk factors of TB transmission to HCWs [5]. 

Several studies have presented the risk factors for the delayed isolation of TB. Commonly suggested risk factors include old age [8,9] and admission to departments other than the pulmonology or infection department [10,11]. Presence of symptoms can be an important indicator for clinicians to suspect TB. Mohammed et al. suggested that early chest X-ray screening in patients with cough minimized diagnostic delay of TB [12,13]. However, few studies have focused on the detailed process of how the clinical characteristics of index patients affect HCWs in terms of in-hospital exposure and transmission.

The National Medical Center (NMC), as the national center for infectious disease with over 300 annually reported TB cases, established and operated the “Tuberculosis Relief Belt Supporting Project” to support patients with TB from vulnerable groups [14,15]. Despite continuous efforts from infection control teams to establish effective infection prevention and control (IPC) measures, a considerable number of unexpected TB exposure events occur annually, causing an increased risk of in-hospital transmission and unnecessary costs. 

With the high prevalence of TB in South Korea, it is difficult for HCWs to avoid unexpected exposure to TB during work. Timely diagnosis of TB is important to protect HCWs. In addition, contact investigation is costly to the healthcare system, because it increases the workload of the infection control team [16]. In this study, we analyzed factors associated with delayed isolation of TB and its impact on in-hospital transmission. We aimed to identify a proper screening protocol and policies to protect HCWs from unnecessary exposure to TB. In addition, by reviewing the contact investigation results at our institution, we suggest an effective infection control strategy to detect TB-infected HCWs with appropriate resources.

## 2. Materials and Methods

### 2.1. Study Participants and Designs

This was a single-center, retrospective study of index patients and close-contact HCWs who were subjected to contact investigation after TB exposure during hospitalization at the NMC (Seoul, South Korea) from January 2018 to July 2021. We reviewed the exposure reports and electronic medical records of index patients and HCWs.

Active TB was diagnosed if any of the following results were present: positive acid-fast bacilli (AFB) culture or Xpert MTB/RIF assay (Cepheid, Sunnyvale, CA, USA), either from sputum or bronchial aspirate. If TB was diagnosed during hospitalization without isolation, the patient was promptly relocated to the isolation unit, and an immediate report to the infection control team was made by the nurse in charge. Each department collects a complete list of HCWs with close contact and sends it to the infection control team, in accordance with the NMC infection control guidelines. The infection control team collects information from the participants on the list through telephone surveys and notifies them to perform chest radiography and interferon gamma-releasing assay (IGRA) at the pulmonology clinic. The test results and the following clinic visits are monitored by the infection control team and are published in exposure reports. The aforementioned information included in exposure reports was thoroughly reviewed by the study.

### 2.2. TB Infection Control Measures in the NMC

After the identification of delayed diagnosis, the infection control team collected a list of HCWs suspected of having close contact with patients with TB without protective equipment. A pulmonologist was consulted to determine the proper range of contact investigation, based on clinical information, such as patient’s infectivity, environment, duration, and the procedure involved. HCWs were defined as those having close contacts with a patient infected by TB for more than eight consecutive or 40 cumulative hours, or those who participated in high-risk, aerosol-generating procedures (such as intubation, bronchoscopy, and suctioning) without protection equipment [17].

It is recommended that close contacts should be performed with chest radiographic and IGRA tests, using QuantiFERON-TB Gold In-Tube assay (Qiagen, Hilden, Germany) two times sequentially, immediately after exposure (baseline) and, then, at 8–10 weeks after exposure. If previous test results existed, chest radiographs within 1 month and IGRA test measurements within 6 months were considered baseline results. When the baseline IGRA result was positive, only a chest radiograph was obtained to rule out active TB. HCWs with a positive IGRA conversion were referred to a pulmonologist to consider active TB or latent TB infection (LTBI) treatment. 

### 2.3. Analysis of the Characteristics of Study Participants

Data were collected from electronic medical records and contact investigation reports of the infection control team. Clinical, radiological, and laboratory data of the patients were obtained, including age, sex, comorbidities, previous TB history, symptoms, admission route, main diagnosis, department of admission, exposure days, AFB smear/culture, Xpert MTB/RIF assay, chest radiography, and/or chest computed tomography. TB exposure data, such as exposure site, route of close contact, HCW occupation, and previous TB history, were reviewed. Data on TB contact investigation included baseline/follow-up chest radiographic and IGRA results, newly diagnosed active TB or LTBI cases, and subsequent treatment.

Delayed isolation was defined as the failure to isolate patients with active TB from the negative-pressure isolation room from the beginning of hospitalization. The exposure day was defined as the period from the time of patient admission to isolation. The radiologic findings of index patients were classified to simplified, suspected impression as follows: pneumonia if consolidation or ground-glass opacity was dominant, active TB if cavitation or tree-in-bud pattern was present, old TB if fibrotic scarring or calcification was present, and lung cancer if a mass or nodule was present.

We devised a flowchart summarizing the processes of delayed isolation of patients with TB, and classified the patients into categories according to the patterns of delayed isolation. We analyzed these categories in-depth to identify the impact of delayed isolation on TB transmission.

### 2.4. Ethical Review

The study was approved by the institutional review board of the National Medical Center (NMC-2021-08-105). The board waived the requirement for informed consent from the patients owing to the retrospective nature of the study. This study was conducted in accordance with the principles of the Declaration of Helsinki.

## 3. Results

### 3.1. Baseline Characteristics of Index Patients

During the study period, among 40 hospitalized patients who were reported to the infection control team to have had delayed isolation, 25 index patients underwent contact investigation. Eight patients did not have a close contact with HCW or inpatients, and seven patients had non-tuberculous mycobacterial infections or false positives. Table 1 shows the baseline characteristics of the 25 patients. Most patients were older (median age, 63 years) and male (88.0%) individuals. Four (16.0%) patients had a history of TB. The median exposure time was 4 (2–12) days. Most (64.0%) patients were admitted via the emergency room. The common symptoms on day 1 were abnormal sputum (44.0%), fever (32.0%), and dyspnea (28.0%). Five (20.0%) patients were asymptomatic. All but one patient showed abnormalities on chest radiography. Twelve (48.0%) patients showed radiologic features suggestive of pneumonia, four (28.0%) had typical findings of active TB, and six (24.0%) presented with old TB. Most (72%) patients were AFB smear-negative and diagnosed with TB based on positive Xpert MTB/RIF assay results. Twice as many patients were admitted to a non-pulmonology/infectious disease department (72.0%) compared with those who were admitted to the pulmonology/infectious disease department (28.0%). The details of TB exposure for each index patient are summarized in Appendix A.

### 3.2. Factors Influencing Delayed Isolation

Figure 1 shows the diagnostic flow for how the index patients experienced delayed isolation. Based on the flowchart, we classified 25 cases into five categories (A–E). 

Category A comprised seven patients who required emergency procedures or surgery upon arrival; thus, clinical information was not available before close contact. Five patients had abnormal radiological findings and were immediately screened for TB in the emergency room. The other two patients, one with pneumonia on chest radiography and the other with respiratory symptoms, were missed in the emergency room and screened during hospitalization. Six of the remaining 18 patients, whose clinical information was reviewed before admission, were admitted to the pulmonology or infection department (categories B and C). Four patients were under preemptive isolation and were released from the unit after checking the negative test results; however, the test result was positive after repetitive sputum AFB smear or bronchial washing, and the patients had to be isolated again (category B). Two patients were neither suspected nor tested for TB, as they had dominant features of other pulmonary diseases (interstitial lung disease and pneumonia) in their radiologic findings (category C).

Among the 12 patients who were admitted to another department but underwent proper clinical review before admission, none showed normal radiologic findings (categories D and E). Four patients were immediately evaluated but were isolated a few days later, because test results were delayed or missed by HCWs (category D). Eight patients were not immediately screened for TB, despite abnormal chest radiographs (category E). Among the eight patients in category E, six had TB-related symptoms on day 1 but were neglected in the non-pulmonology or infectious disease department (Appendix A). Two of them were detected after the diagnosis of extrapulmonary TB, and four of them were diagnosed during the evaluation of hospital-acquired pneumonia. The other two patients were found during evaluations for other purposes, such as organ donor screening and cancer workup.

### 3.3. Features of Close-Contact HCWs According to the Type of Delayed Isolation

A total of 125 HCWs were investigated because of their close contact with the 25 index patients. The infection control team reported 157 exposure events. Details on the exposure events that occurred for each index patient and category are summarized in Appendix A. Exposure duration was the shortest in category A (median 1 day) and the longest in category E (median, 13.0 days), in which patients were not immediately evaluated despite an abnormal chest radiograph. The median number of exposure events per index patient was the greatest in category A patients (*n* = 8), followed by category E patients (*n* = 6.5). There was no close contact between HCWs and the two index patients in category C.

Twenty-five (20.0%) HCWs had more than one close contact with different index patients (57 exposure events). Table 2 shows the information on multiple-contact HCWs and the situation of exposure. In terms of occupation, 16 (64.0%) were nurses who had been in contact with the patient for more than 40 h or participated in aerosol-generating procedures; eight (32.0%) were doctors, all of whom were exposed during intubation; and one was a pulmonary function test technician. In terms of the department, 10 (40.0%) HCWs belonged to the emergency department and the other 10 (40.0%) were from the intensive care unit. Exposure events in multiple-contact HCWs were most frequent in category A (*n* = 37, 64.9%), accounting for over half of the total multiple-contact events, followed by categories E (*n* = 10, 17.5%) and B (*n* = 9, 15.8%).

### 3.4. Results of Contact Investigation in HCWs

Among the 157 exposure events involving 125 HCWs, a complete schedule of screening tests was performed for 128 events (97 HCWs). Four HCWs (one HCW in two events) were pregnant, 18 refused to undergo follow-up examinations, and six resigned from the hospital during the follow-up period. Among the 97 HCWs, 14 had a history of TB. Moreover, 19 HCWs had close contact with AFB smear-positive index patients (seven patients in total; Table 1 and Appendix A), but no positive conversion of IGRA occurred. Of the 83 TB-naïve HCWs, only one (1.2%) presented a positive conversion of IGRA after the exposure event (HCW no. 2 in Table 2). HCW no. 2 was an emergency medicine resident who participated in the intubation of Index Patient no. 20. He received LTBI treatment (isoniazid and rifampin) for 3 months. Unfortunately, shortly after treatment initiation, he had two additional close contacts with index patient no. 2 and 4, which also occurred during intubation. None of the HCWs were diagnosed with active TB after 8–10 weeks of observation, and there was no voluntary notification after the observation period.

## 4. Discussion

In this study, we identified 25 TB index cases, in which delayed isolation occurred, resulting in 157 exposure events in 125 HCWs. Except for emergency procedures performed without clinical information (category A) or a positive AFB smear after repeat testing with the impression of TB (category B), delayed isolation occurred when the clinician neglected abnormal radiologic findings or related symptoms. Close-contact events occurred mainly in category A (pre-admission in the emergency room), and most HCWs with multiple contacts and those diagnosed with LTBI were also included in this category.

The clinical characteristics of our index patients were consistent with those reported in other studies, such as admission to departments other than pulmonology or infection [8,10,11], negative AFB smear [3], absence of cavity [3,18], and concurrent pulmonary parenchymal disease in radiologic findings [19].

To avoid nosocomial transmission, routine chest radiograph screening could be suggested in high-TB burden settings before or immediately after admission [20]. Despite most index patients presenting with abnormal radiologic findings in our study, many were not initially suspected of having TB. Radiological reports are often considered insignificant or missed by clinicians. As TB often presents with atypical chest radiographs, patients admitted to the non-pulmonology or infectious disease department are more susceptible to isolation delay [11]. Thus, for patients in non-pulmonology or infectious disease departments, clinicians should not be reluctant to consult pulmonologists for abnormal chest radiographs. In the case of hospitals with a higher incidence, more active suspicion of TB and additional tests should be considered. To prevent cases caused by neglected radiology reports, systematic improvements, such as critical test result notification, can provide an effective solution [21,22].

Our results suggest that TB-related symptoms are often neglected in clinical settings, especially in patients hospitalized in non-pulmonary or infectious disease departments. TB should be suspected if the patient presents with relevant symptoms, such as fever or respiratory symptoms persisting for weeks, or features of other respiratory diseases [6]. However, in our study, six of eight patients in category E initially had TB-related symptoms, but did not lead to early detection. An American study found that a considerable number of patients missed this opportunity before TB diagnosis: during the so-called “diagnostic opportunity window,” these patients either showed symptoms of TB or were diagnosed with a symptomatically similar disease [23]. Considering the higher prevalence of TB in South Korea, clinicians should be careful to avoid cognitive bias and always consider the possibility of TB. The South Korean government promotes campaigns for TB prevention through media to teach proper coughing etiquette, and encourages physicians to check screening tests if a cough lasts for more than 2 weeks [24,25]. 

The largest number of close-contact events occurred in category A, despite the shortest exposure period. The diagnosis of TB can easily be missed or delayed in the emergency department, leading to repetitive exposure of HCWs [26]. Hefferman et al. reported that patients with TB who visited the emergency department were likely to have longer nosocomial exposure times and more secondary cases [27]. Personal protective equipment, including particulate respirators, provides additional protection, where administrative and environmental controls cannot completely prevent TB transmission [28]. In the current guidelines, particulate respirators are recommended for HCWs treating TB suspected or confirmed TB [29]. However, for category A patients in our study, many patients did not have time to be fully assessed before the procedure. Our study results suggest that in institutions with a high TB burden, recommending protective equipment during emergency procedures for all patients without information may help reduce HCW transmission. In contrast, no close-contact events occurred in category C patients. HCWs avoided unexpected exposure because the patients were admitted to the ward and did not require an aerosol-generating procedure. This is in line with the findings of Kim et al., who reported that more patients with TB in the intensive care unit underwent high-risk procedures than those in the general ward [30]. Thus, during admission, respirators should be considered for high-risk situations, such as cough-inducing and aerosol-generating procedures [28].

Interestingly, in our study, there was only one (1.2%) positive conversion of IGRA out of 83 TB-naïve HCWs after exposure events; the impact of index patients was minimal compared with that reported in other studies. In another South Korean study, contact investigation involving 157 HCWs from 206 exposure events of 24 index patients, LTBI developed in nine (9.2%) of 98 TB-naïve HCWs. Common procedures were similar to those reported in our study, such as respiratory care (suction) and nursing for a long period [26]. In another South Korean study of 872 HCWs exposed to 55 index cases, 10 (2.4%) of 415 HCWs had tuberculin skin test conversion, and one (0.2%) HCW developed active pulmonary TB [31]. The relatively low number of patients in our study can be explained by the effective administrative control of the NMC. Our results suggest that infection control protocols can significantly reduce in-hospital transmission of TB. 

According to WHO guidelines, TB IPC protocols should be based on a three-level hierarchy: administrative, environmental, and respiratory protection [32]. The first and most important level of all IPC programs is administrative control [33]. At our institution, it is recommended to check the Xpert MTB/RIF assay in patients with pneumonia or a history of TB. If the test result is positive, it is immediately reported to the ward and clinician in the form of a critical value report. As this practice significantly improved the detection rate, it can be suggested that the index patients in our study—of whom only 28% had a positive AFB smear—showed low infectivity and caused fewer in-hospital transmissions. Our practice is in line with findings from other South Korean studies supporting the Xpert MTB/RIF assay for timely diagnosis [4] and as a screening test, especially for patients with pneumonia [20]. 

Our study had a few limitations. First, it was a retrospective study with a small sample size; therefore, statistical analysis could not be performed. Second, there may be more index cases that have not been identified. Finally, after the COVID-19 outbreak, the total number of inpatients with TB declined and the composition of the patient population changed. In addition, the frequency of wearing personal protective equipment in HCWs has increased; thus, the impact on index patients might have been underestimated. 

## 5. Conclusions

In conclusion, 25 index patients were reported to have delayed isolation with HCW exposure, leading to 157 exposure events among 125 HCWs. Among 83 TB-naïve HCWs who completed the contact investigation, only one (1.2%) from category A was diagnosed with LTBI after exposure. Delayed isolation and TB exposure mostly occurred in the pre-admission stage, during emergency situations and aerosol-generating procedures. Effective TB screening and infection control protocols for TB are necessary for HCWs to contact new patients in high-risk departments.

## Figures and Tables

**Figure 1 jcm-12-01361-f001:**
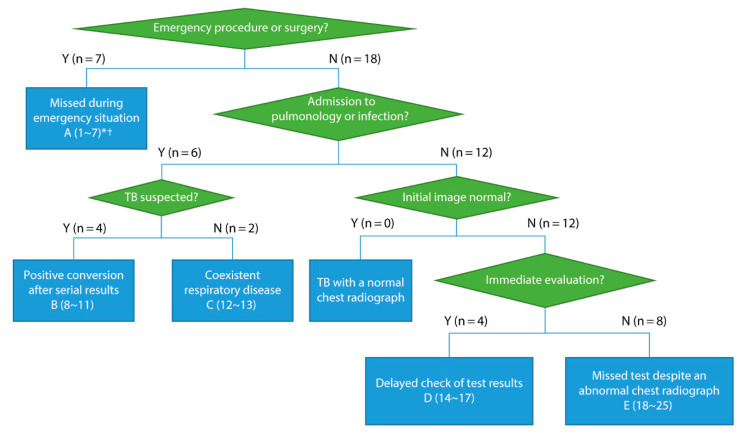
Diagnostic flow of index patients. * Category (index number). TB, tuberculosis. † One patient in category A was admitted to the pulmonology department.

**Table 1 jcm-12-01361-t001:** Baseline characteristics of index patients (N = 25).

Age, median (IQR)	63 (54–71)
Sex, men	22 (88.0%)
**Underlying diseases**	
Hypertension	10 (40.0%)
Diabetes	5 (20.0%)
Malignancy	4 (16.0%)
Chronic pulmonary disease	3 (12.0%)
Chronic liver disease	2 (8.0%)
Chronic kidney disease	1 (4.0%)
Coronary artery disease	0 (0.0%)
HIV infection	0 (0.0%)
**Previous history of TB**	4 (16.0%)
**Symptoms**	
None	5 (20.0%)
Abnormal sputum *	11 (44.0%)
Fever	8 (32.0%)
Dyspnea	7 (28.0%)
Mental change	6 (24.0%)
Cough	5 (20.0%)
Anorexia	5 (20.0%)
General weakness	4 (16.0%)
Chill	3 (12.0%)
Chest pain	1 (4.0%)
Hemoptysis	0 (0.0%)
Night sweats	0 (0.0%)
**Initial radiologic findings**	
Pneumonia	12 (48.0%)
Active tuberculosis	4 (16.0%)
Old tuberculosis	6 (24.0%)
Lung cancer	2 (8.0%)
Normal	1 (4.0%)
**Bacteriological examinations**	
AFB smear negative	18 (72.0%)
AFB smear positive	7 (28.0%)
1+	3 (12.0%)
2+	2 (8.0%)
3+	2 (8.0%)
MTB culture positive	16 (64.0%)
Xpert MTB/RIF assay positive	23 (92.0%)
**Extrapulmonary TB**	2 (8.0%)
**Hospitalization course**	
Route of admission	
Emergency room	16 (64.0%)
Outpatient	9 (36.0%)
Exposure days, median (IQR)	4 (2–12)
**Department of admission**	
Pulmonology	7 (28.0%)
Other department	18 (72.0%)
Gastroenterology	3 (12.0%)
Neurology	3 (12.0%)
General surgery	2 (8.0%)
Nephrology	2 (8.0%)
Neurosurgery	2 (8.0%)
Orthopedic surgery	2 (8.0%)
Hematology & oncology	1 (4.0%)
Ophthalmology	1 (4.0%)
Thoracic surgery	1 (4.0%)
Urology	1 (4.0%)

HIV, human immunodeficiency virus; TB, tuberculosis; AFB, acid-fast bacilli; IQR, interquartile range; MTB, *Mycobacterium tuberculosis*. * Abnormal amount, color, or odor of sputum.

**Table 2 jcm-12-01361-t002:** Healthcare workers with multiple contacts.

HCW No.	Occupation	No. of Close Contact	Index Patient	Category (No. of Patients)	Department of Index Patient	Exposure Site	Route of Close Contact
1	Emergency physician (resident)	3	1, 3, 5	A (3)	Neurology, Orthopedic surgery, Pulmonology	ER	intubation
2	Emergency physician (resident)	3	2, 4, 20	A (2), E (1)	Neurology (2), Neurosurgery	ER, ICU	intubation
3	ER nurse	3	2, 3, 5	A (3)	Neurosurgery, Orthopedic surgery, Pulmonology	ER	intubation
4	ER nurse	3	2, 4, 6	A (3)	Neurosurgery, Neurology Gastroenterology	ER	intubation
5	ICU nurse	3	4, 6, 9	A (2), B (1)	Neurology, Gastroenterology, Pulmonology	ICU	intubation, suction, contact time over 40 h
6	ICU nurse	3	4, 6, 24	A (2), E (1)	Neurology, Gastroenterology (2)	ICU	suction, contact time over 40 h, EGD
7	ICU nurse	3	4, 9, 24	A (1), B (1), E (1)	Neurology, Pulmonology, Gastroenterology	ICU	suction, contact time over 40 h
8	Emergency physician (attending)	2	2, 20	A (1), E (1)	Neurosurgery, Neurology	ER, ICU	intubation
9	Emergency physician (resident)	2	3, 5	A (2)	Orthopedic surgery, Pulmonology	ER	intubation
10	Emergency physician (resident)	2	2, 5	A (2)	Neurosurgery, Pulmonology	ER	intubation
11	Anesthesiologist (resident)	2	7, 23	A (1), E (1)	General surgery, Thoracic surgery	OR	surgery
12	Neurologist (resident)	2	1, 20	A (1), E (1)	Neurology (2)	ER, ICU	intubation
13	Doctor(intern)	2	5, 9	A (1), B (1)	Pulmonology (2)	ER, Ward	intubation, contact time over 40 h
14	ER nurse	2	1, 5	A (2)	Neurology, Pulmonology	ER	intubation
15	ER nurse	2	2,5	A (2)	Neurosurgery, Pulmonology	ER	intubation
16	ICU nurse	2	4, 6	A (2)	Neurology, Gastroenterology	ICU	suction, contact time over 40 h
17	ICU nurse	2	4, 6	A (2)	Neurology, Gastroenterology	ICU	suction, contact time over 40 h
18	ICU nurse	2	4, 9	A (1), B (1)	Neurology, Pulmonology	ICU	suction, contact time over 40 h
19	ICU nurse	2	4, 9	A (1), B (1)	Neurology, Pulmonology	ICU	suction, contact time over 40 h
20	ICU nurse	2	4, 24	A (1), E (1)	Neurology, Gastroenterology	ICU	suction, EGD
21	ICU nurse	2	7, 22	A (1), E (1)	General surgery, Neurosurgery	ICU	suction
22	ICU nurse	2	7, 22	A (1), E (1)	General surgery, Neurosurgery	ICU	suction
23	Ward nurse	2	9, 10	B (2)	Pulmonology (2)	Ward	suction
24	Ward nurse	2	9, 10	B (2)	Pulmonology (2)	Ward	suction
25	PFT technician	2	17, 25	D (1), E (1)	Gastroenterology, Urology	PFT unit	PFT

ER, emergency room; ICU, intensive care unit; OR, operation room; PFT, pulmonary function test; EGD, esophagogastroduodenoscopy; HCW, healthcare worker.

## Data Availability

The data that support the findings of this study are available from the corresponding author, upon reasonable request.

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
