# Peer review of "Predictive Factors and Clinical Impacts of Delayed Isolation of Tuberculosis during Hospital Admission"

_jcm, 2023, doi:10.3390/jcm12041361_

Round 1

Reviewer 1 Report

I thanks the authors for doing this important TB transmission control study 

My general recommendation is 

1. Better if they add to introduction the purpose of early screening for TB through symptoms and/or chest X-ray from existing evidences among other: 

 1. DOI: 10.4103/ijmy.ijmy_216_21

2. DOI: 10.2147/RMHP.S337392

2. Better if appropriate terms used for e.g. "study subjects" on sub titles under 2.1 and 2.3, better if "study participants" was used 

3. Better if the methods part were improved, e.g. from the data how did the authors determined the close contacts professions from different   departments with the frequency of contacts? What was the data sources, the number of tests and types of testes that were performed for HCWs were unclear  

4. What is "sputum" under TB symptoms? Sputum is sample, not symptom

5. Those diagnosed at pulmonology was 6, not 7 as indicated in Table 1

6. What written under 3.4 part of results needs edition 

..."He was an emergency medicine resident who.."  He has to be changed HCW No.6, it was identifier! 

Reviewer 2 Report

The manuscript in well-written. Results are compiled in  very comprehensive way.

1. My only concern is that about the retrospective nature of this study. The topic chosen for this study needs a prospective design of study for better results. As mentioned by the author himself, retrospective studies would have many drawbacks for enrolling the patients and also getting exact exact representation of clinical impacts. 

2. The author has commented that there were 7 AFB positive index patients (3, 1+ ; and 2 each of 2+ and 3+). Has the author tried to relate the exposures of these index cases because they represent open cases and chances of exposure to other persons are high after open cases.  If the data is there, author should comment on it in result section.
